# ColorArt: Suggesting Colorizations for Graphic Arts Using Optimal Color-Graph Matching

Murtuza Bohra
murtuza.bohra@research.iiit.ac.in

Vineet Gandhi
vgandhi@iiit.ac.in

Center for Visual Information Technology, KCIS, IIIT-Hyderabad

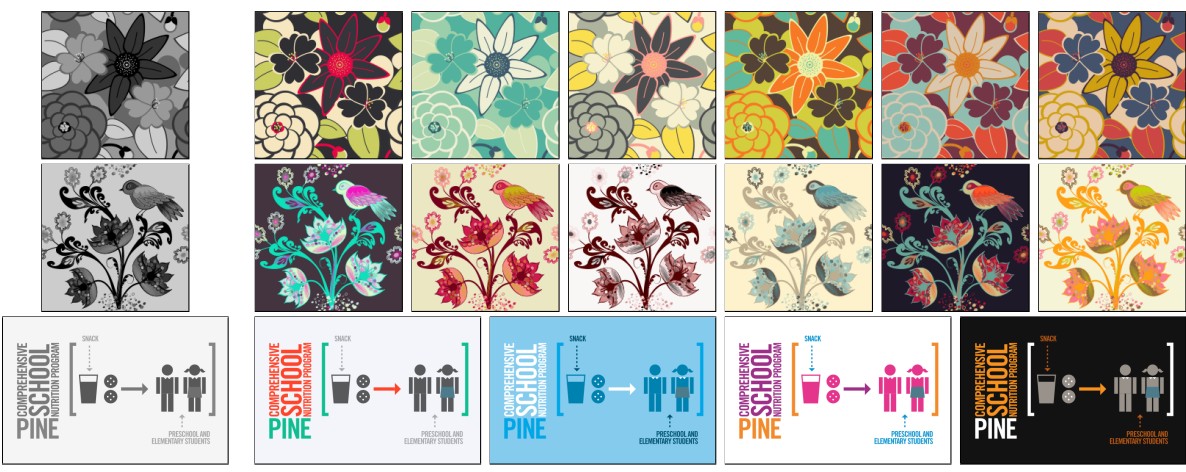

Template          Color Suggestions

Figure 1: For gray-scale template on the left, coloring suggestions using the proposed colorization algorithm are shown on the right.

## ABSTRACT

Colorization is a complex task of selecting a combination of colors and arriving at an appropriate spatial arrangement of the colors in an image. In this paper, we propose a novel approach for automatic colorization of graphic arts like graphic patterns, info-graphics and cartoons. Our approach uses the artist's colored graphics as a reference to color a template image. We also propose a retrieval system for selecting a relevant reference image corresponding to the given template from a dataset of reference images colored by different artists. Finally, we formulate the problem of colorization as a optimal graph matching problem over color groups in the reference and the template image. We demonstrate results on a variety of coloring tasks and evaluate our model through multiple perceptual studies. The studies show that the results generated through our model are significantly preferred by the participants over other automatic colorization methods.

**Index Terms:** Graphic-arts, Pattern colorization, Infographics, Color Graph Matching, Automatic colorization, Reference based colorization

## 1 INTRODUCTION

In this work, we study the problem of automated colorization of gray-scale templates (Figure 1). The two key sub-problems in colorization are: (a) guess what colors look aesthetically pleasing when used together and (b) the spatial arrangement of them in a given template. It is considered a valuable ingredient of our artistic abilities, so and so, that children are introduced to colors and color-books at an age, as early as 2-3 years. After years of exposure and practice,

most of us can reasonably gauge whether a color pattern looks aesthetically pleasing or not, and use that knowledge in day to day lives to make color choices on things like clothes, furniture, paint or wallpapers. However, creating attractive colored patterns from scratch is a challenging task and is often left to the artists. Even artists employ a "guess and check" approach for colorization, which involves many trials before reaching the final result. Hence an algorithm that can automatically suggest different colorizations for a given template would be handy for both the regular users and the artists. In this paper, we propose a novel way to generate coloring suggestions for a given gray-scale graphic template (illustration in Figure 1). The proposed solution is generic and caters to variety of applications like alternate coloring suggestions for webpage designs, patterns, clip art, fashion designs, or infographics.

Several studies have looked at the problem of the automated colorization of gray-scale patterns. Notable efforts have been made to build computational models to quantitatively measure the aesthetics of a colored pattern. Such models either try to model psycho-visual observations [13, 21] or take a data-driven approach [18] which learns regression functions from a database of palettes and corresponding user ratings. Different colorizations are then evaluated and selected based on the score from these aesthetics prediction models. For instance, given a template and a palette, the work by Kim [10] evaluates all permutations of the possible coloring and chooses the permutation leading to the highest score in term of aesthetics. However, the palette selection remains a challenge, and also an exhaustive search is computationally prohibitive (for a pattern with ten colors, we would need to evaluate over 3.6 million permutations). Another notable direction is to build data-driven generative models for direct colorization. A pioneering effort in this space came from Lin [14], who build factor graphs by learning feature-based unary and binary potentials, combined with a compatibility cost [18]. However, such models are not intuitive; they are computationally heavy and give little control to the user for interactivity.

Graphics Interface Conference 2020
28-29 May

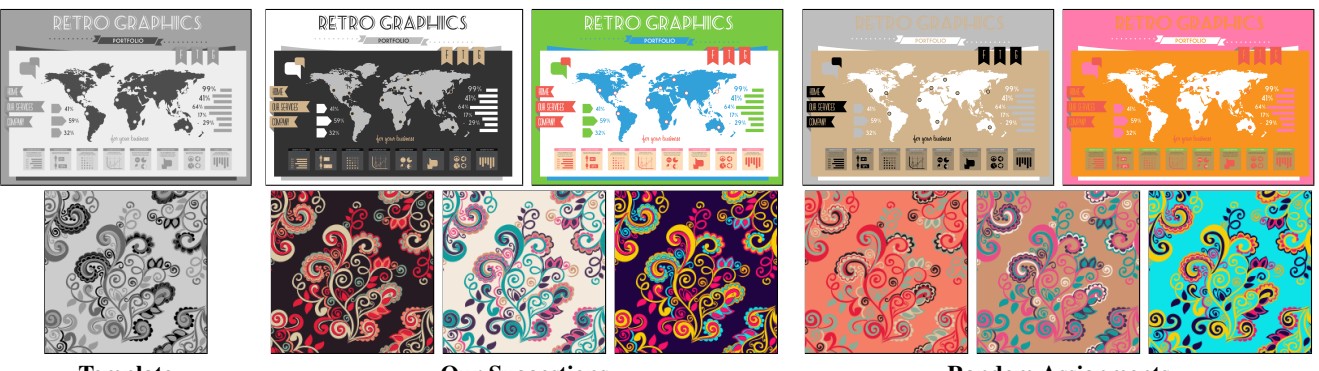

| **Template** | **Our Suggestions** | **Random Assignments** |

Figure 2: Template images(left); Our coloring suggestions(middle) and Random assignment(right), shows the aesthetic importance of appropriate spatial arrangement of colors. Corresponding images in middle and right column uses the same palette but different color assignments.

This paper proposes a novel two-stage framework for pattern colorization. In the spirit of previous work, our approach is also data-driven and relies on the large dataset of artist colored graphics. Given a gray-scale template, the first stage of our algorithm retrieves appropriate images from the artist's database to be used as a reference. The images from the dataset which are closer in color composition and texture distribution are termed more appropriate for coloring the given template. The second stage optimizes the color assignments i.e., propagating colors from the retrieved reference image to the input template. If the template and reference images have $N$ distinct color groups (each color groups pertaining to a specific color or gray-scale value), then $N!$ color assignments are possible. The aim is to search for an optimal assignment (Figure 2 contrasts random assignments vs optimal ones).

We hypothesize that two important aspects parameterize the color intent of an artist: (a) The composition of colors i.e., the proportion in which different colors are used. Some colors cover large parts of the screen to create a subtle impression (mood), while some are used only on a small portion to make things stand out (like flowers or warning signs). (b) The adjacency of colors i.e., what colors are used around each other. The adjacency controls aspects like separation, contrast, etc. Consequently, we pose the assignment as a weighted graph matching problem (WGMP) [1] between the color graphs (where each vertex corresponds to a color group, and edge weight defines the number of adjacent pixels between two color groups) of the reference and the template image. The optimization is to preserve the spatial color adjacency and color compositions of the reference image onto the target image. We use an analytic approach, instead of a combinatorial or an iterative one to obtain the graph matching solution in polynomial time, which makes our approach computationally efficient and scalable in terms of the number of distinct color groups. Although approximate, analytic solutions are shown to find a matching close to the optimum one, especially when the graphs are sufficiently close to each other. We evaluate the colorings generated by our algorithm using judgment studies and demonstrate that they were significantly preferred over the existing state of the art algorithms.

More formally, we make following contributions:

1. We propose a reference image retrieval system based on spatial features of the template image.

2. A novel color transfer algorithm to color a gray scale image from a reference image based on color graph and composition matching.

3. Thorough judgement studies to compare our proposed automatic image colorization system against the artist versions and the state of the art algorithms.

## 2 RELATED WORK

### 2.1 Color Palette Selection

Automatic colorization involves palette selection and finding an appropriate spatial arrangement of these colors in the template image. Most of the previous literature tackles these two problems independently. For palette selection, early efforts of finding compatible colors date back to the 17th Century when Newton presented the color wheel and drew analogies between harmonies of sounds and harmonies of colors. In 18th Century, Goethe in his book [4], proposed that compatible contrasting colors are opposite on the color wheel. One of the most popular theories of color compatibility is the notion of hue templates [8, 9], which generalizes Goethe's theory by describing compatible colors as fixed rotations about the color wheel. However, designers often treat templates only as starting points, rather than strict rules [16].

Ishibashi and Miyata [6, 7] proposed an interactive system to search the user's desired color scheme. They represent a color palette using statistical features and sample colors by fitting a model over the entire dataset. The sampling is conditioned over user preferences. There are numerous researches on measuring color harmony [2]. Ou [13] performed psychophysical experiments to build a computational model, to predict color harmony between a pair of colors. Zhang [24] studied the correlation of color harmony with shape & layout and demonstrated that harmony predictions are incomplete without considering shape & layout information.

An alternate line of work for palette selection takes a data-driven approach. O'Donovan [18] curated a large dataset of color palettes and their user rating from color websites; ColourLovers [1] Kuler [2]. They further employed MTurk to get additional user ratings for 10,743 color palettes. They attempt to approximate human color preferences by regression model over 334 dimensions feature vector(features from different color spaces) from each color palette to predict user ratings. To overcome the limitation in [18], which uses fix sized color palettes, Kita [11] extracts features independent of the number of colors in the palette. They learn a regression model to predict the overall compatibility score for an arbitrarily large color palette.

Recent work by Mellado [17] on palette space exploration highlights the dependence of palette on the content of the image. Need for human interaction to arrive at an appropriate palette also suggest that finding a good palette is not an independent problem from automatic coloring an image. O'Donovan's large scale studies also confirm that judging only the palette (especially with more colors) is difficult than judging colors in an image (evident from high varia-

---

[1]https://www.colourlovers.com/
[2]https://color.adobe.com

tion in user ratings for color schemes). Consequently, in this work, we couple the problem of palette retrieval with the content (spatial features) of the input image (Section 4.1).

## 2.2 Image Colorization

Even when the color palette is given, image colorization is challenging because the different spatial assignments of color lead to the different overall appearance of the resultant image. Mainly the color appearance (desired soothing & harmonious effect) depends up on the selection of colors for spatially neighbouring segments. Sartori [20] studied how different color combinations used together in a painting affect viewers emotionally.

Zhang [23] proposed an approach to recolor images with a given palette. They first extract the dominating colors from the image and find the linear decomposition of each pixel in extracted palette. Then using linear coefficients and different permutations of given palette they generate different coloring results. However, their algorithm does not suggest which permutation gives better coloring. Kim [10] proposed another palette based colorization approach by leveraging computational models for harmony and contrast. They perform an exhaustive search and evaluate all possible colorization of the template using the given palette. Their algorithm is limited by the reliance on the user to provide the palette and is also computationally prohibitive if the number of distinct colors are large. We pose the color assignment problem as a color transfer from the reference to target image using weighted graph matching, whose best approximate solution is in $\mathscr{O}(N^3)$ (in contrast to $N!$ in [10]).

There have been attempts for fully automatic colorization of graphic arts without explicitly requiring a palette. Lin [14], employed factors graphs to learn discrete probability distribution functions from a representative set of pairs of palettes and images. The complexity and lack of adaptability of frozen models (post-learning) limit the user interactivity. Furthermore, their perceptual studies find gap in the aesthetic appeal of the predicted models in contrast to artist's work. This serves as one of the baselines for us, where we attempt to bridge the gap between automatic colorization of a gray-scale template with it's colored versions produced by different top artists.

Colorization of graphic arts is different from coloring natural images because the contents in natural images have high semantic meaning and often colors are associated with semantics e.g. sky is always blue, grass are green and so on. Graphic arts has lesser semantic constraints and that leads to more choice of colors. Our work however takes inspiration from the literature of automated natural image colorization. Initial approaches [12, 15] colorize a gray-scale image based on user-guided scribbles, which requires professional skill to provide good scribbles. To overcome the dependence on user, Zhang [25] proposed a fully automatic colorization using convolution neural networks trained in a self-supervised way (using a large dataset of colored images and corresponding grayscale images). However, this approach does not allow multiple coloring suggestion for same image, which is desired in graphics domain studied in this work. An alternate line of thought is to use a sample reference image for colorization [3, 19]. Recent methods show impressive colorization results [5] using exemplar image as color source. Taking inspirations from these, we also propose an exemplar-based colorization algorithm for graphic arts.

## 3 REFERENCE IMAGES DATASET

To build our colored reference image dataset (CRID), we use a set of colored patterns collected from Colourlovers (an online community centered around creating and sharing color designs) in [14]. We further augment the dataset it with recent top-ranked design patterns from ColourLovers. To demonstrate the applicability of our approach in a variety of coloring tasks, we also created a separate Vector Graphic (VG) dataset by collecting a set of infographics,

cartoons, and animations by crawling images from the web. We only collected vector images for precise palette extraction and avoid compressed formats, as they lead to false colors.

Our CRID dataset contains total of 8258 patterns, each pattern colored with 5 distinct colors. We reuse 7258 colored patterns previously collected by Lin [14] and added 1000 recent top-ranked patterns from colorLovers. The dataset has 3727 unique design templates, among these, some templates have multiple instances with different colorization by multiple artists. Our VG dataset contains 1514 cartoon and animated scenes and 300 infographics images. We pre-process images in the dataset to represent each image as color groups, which will be later used in our retrieval and color transfer algorithm. We first assign each pixel in the image to one of the colors in the source palette. We then group pixels corresponding to each color as connected components(with size greater than 15 pixels to account for noise), which are 2 pixels or closer to each other and call them 'segments'. Finally, we group all the segments with the same color into the corresponding color group.

## 4 METHODOLOGY

The overall approach is illustrated in Figure 3. Input to our method is an image $I_{Temp}$, which is a grayscale pattern with $N$ distinct lightness values. We refer to the segments that map to the same lightness value as a color group. Our algorithm aims to replace each distinct lightness value by a different colors from a retrieved reference image $I_{Ref}$ to generate colored output image. To scope this work, our algorithm requires both, the template and the reference images to have equal number of color groups. The following sections describes our algorithm in two stages; a) Retrieval of reference image, and b) Use the reference image to produce coloring suggestion for input template image.

## 4.1 Reference Image Retrieval

The aim of the retrieval algorithm is to find images from the CRID dataset, suitable for input template. Previous work by Lin [14] has shown that color choice for a group correlates with its geometrical properties (like relative size, spread, number of distinct segments, etc.). For instance, large and highly connected segments usually represent the background. The presence of such a large segment indicates that the artist intends to use a background, which is often so chosen that it brings the focus on the foreground (loud colors in the background are less preferred). On the other hand, if all color groups are equally distributed spatially, then the pattern is not intended to have a background foreground separation. Therefore, we hypothesize that reference images with similar geometrical properties would be a better choice to color a given gray-scale template. Consequently, we perform feature extraction to represent the geometrical properties of each color group in the given image and concatenate them to represent the entire image. Then, reference images are selected from the dataset using a K-Nearest-Neighbour (KNN) algorithm in the represented feature space ($K$ can be selected based on number of coloring suggestions to be generated). In case of VG dataset, which has images with different number of color groups, we first filter the images having the same number of color groups as template image before searching for reference image. We use following spatial features:

**Composition Vector** (*C*): Each element in the composition vector represent size of corresponding color group in the image i.e. $C = [c_1 c_2 ... c_p]$, where $c_i$ is the ratio of the number of pixels in $i^{th}$ color group to the total pixels in the image, and $p$ is total color groups in the image. The groups are arranged in descending order of sizes with $c_1$ being the largest group.

**Spread of Color Groups** (*S*): This feature captures the distribution of segments in a color group i.e. is it a single large connected component, or it consists of several disjoint connected components.

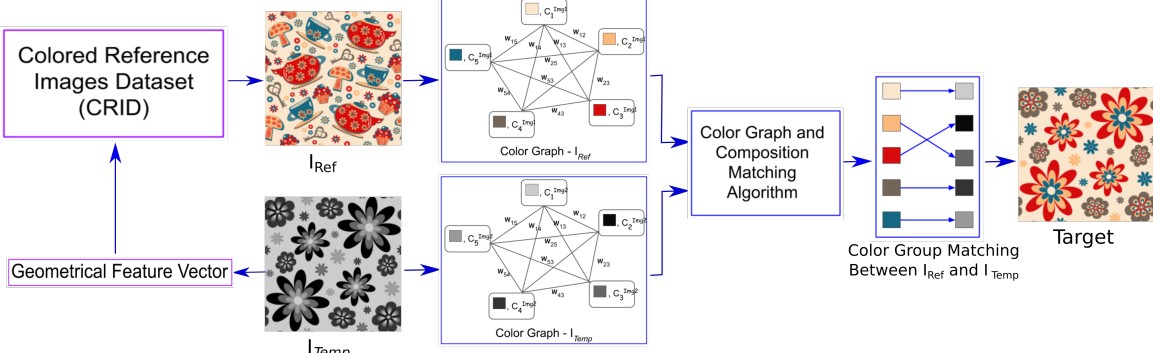

Figure 3: Block diagram represents overall proposed methodology. $I_{Temp}$ in the figure is template image and $I_{Ref}$ is the reference image found closest to $I_{Temp}$ in the feature space. Each node of both the color graphs represents 'color' and 'composition' of the corresponding color group. Output is the target image which is produced by transferring the colors from $I_{Ref}$ to $I_{Temp}$.

We define $S = [n_1 n_2 ... n_p s_1 s_2 ... s_p]$, where $n_i$ is the number of segments which belong to the $i^{th}$ color group divided by the total number of distinct segments and $s_i$ is the average size of segments in $i^{th}$ color group normalized with the size of the image.

**Sum of Gradient** $(m)$**:** We also compute the sum of the gradient magnitude of the entire image, as a measure of overall texture. Overall we define:

$$G_I = [c_1^I c_2^I ... c_p^I n_1^I n_2^I ... n_p^I s_1^I s_2^I ... s_p^I m^I]_{1 \times 3p+1}. \qquad (1)$$

Where $G_I$ is the spatial feature vector representing image $I$ and, $c_i^I, n_i^I, s_i^I$ and $m^I$ are composition, number of segments, average size of segments corresponding to $i^{th}$ color group and the sum of gradient magnitude for image $I$ respectively. Therefore, distance $D$ between two images $I_1$ and $I_2$ to find nearest neighbour, is defined as euclidean distance between $G_{I1}$ and $G_{I2}$ i.e. $D = ||G_{I1} - G_{I2}||$.

### 4.2 Color Transfer

The objective color transfer algorithm is to propagate colors of reference image to the input template, preserving the original color intent of the artist in the reference image. The intent of the artist is defined in terms of composition of different colors used by artist and spatial adjacency between these colors in the image. We formulate the color transfer problem as one to one mapping between color groups of reference and template image. There exist $N!$ such mappings, and the goal is to solve for the optimum one i.e. the mapping with minimum loss of color's composition and adjacency in the output image with respect to the reference image. We solve this problem as optimal color graph and composition matching. We define *Color Graph Matching* and *Composition Matching* below, and subsequently combine them to form a overall matching cost.

**Construction of Color Graph:** Given an image and its color groups, we create a weighted color graph $G = (V, W)$ where $V$ is the set of vertices and $W$ is the weighted adjacency matrix. Each node in the graph $G$ corresponds to a color group in the image; therefore, $|V| = p$, where $p$ is total color groups in the image. And the element $w_{ij} \in W, \forall i, j \in \{1, 2...p\}$ is the edge weight between vertices $v_i, v_j \in V$, where $w_{ij}$ is the count of edge pixels shared between color group $i$ and $j$, normalized with total edge pixels in the image.

**Color Graph Matching:** Let $R = (V_R, W_R)$, $T = (V_T, W_T)$ be color graphs of reference and template image respectively. Here $v_i^R \in V_R$ and $v_i^T \in V_T$ represents nodes in the graph $R$ and $T$. Adjacency of the two graphs are preserved using Weighted Graph Matching Problem (WGMP). The aim of WGMP is to find a one-to-one correspondence $\phi$ between set $V_T$ and $V_R$, which minimizes the

difference between graph $R$ and $T$. The difference is measured as:

$$J(\phi) = \sum_{i=1}^{n} \sum_{j=1}^{n} [W_T(v_i^T, v_j^T) - W_R(\phi(v_i^T), \phi(v_j^T))]^2 \qquad (2)$$

If $P$ is a permutation matrix then above equation can be written in matrix form as:

$$J(P) = ||W_T - PW_R P^T||^2 \qquad (3)$$

where the permutation matrix $P$ represents the node correspondence $\phi$ and $||.||$ is the Euclidean norm ($||W|| = (\sum_{i=1}^{n} \sum_{i=1}^{n} |w_{ij}|^2)^{1/2}$). Thus the WGMP reduced to problem of finding permutation matrix $P$ which minimizes $J(P)$. For isomorphic graphs, we can find permutation $P$ such that $J(P) = 0$. This may happen in specific cases, like coloring a gray-scale pattern using the same colored pattern.

Since finding the exact solution for WGMP is a purely combinatorial and requires $\mathcal{O}(p!)$, we use an Eigen decomposition-based approximate solution [22] in $\mathcal{O}(p^3)$. Which solves *WGMP* as finding a permutation $P$, which maximizes the following:

$$\arg\max_{P} \{tr(P^T \overline{U}_R \overline{U}_T^T)\} \qquad (4)$$

where $tr(.)$ is the trace of the matrix, $\overline{U}_R$ and $\overline{U}_T$ are the matrices whoes elements are equal to the absolute value of corresponding element in $U_R$(eigen vector matrix of $W_R$) and $U_T$(eigen vector matrix of $W_T$) respectively. e.g. $\overline{u}_{ij}^R = |u_{ij}^R|$ where $\overline{u}_{ij}^R \in \overline{U}_R$ and $u_{ij}^R \in U_R$. It can be solved directly using Hungarian matching.

**Composition Matching** $(M_{cmp})$**:** This is to preserve the color composition in the reference image when colors are propagated to the template image. This aims to retain the original intent of the artist e.g. background, may likely remain as a background color (both being the largest color groups). Given the reference image $I_R$ and the template $I_T$ with $p$ color groups each, and corresponding composition vectors $(C_{I_T}, C_{I_R})$. The composition matching cost is defined as follows:

$$c_{ij} = |c_i^{I_T} - c_j^{I_R}|$$

Where, $c_i^{I_T} \in C_{I_T}$, $c_j^{I_R} \in C_{I_R}$, and $c_{ij}$ is the cost of matching the composition of $i_{th}$ color group in template to the $j_{th}$ color group in the reference image. This pairwise matching cost is integrated in the Hungarian matching framework and solved jointly with the *WGMP* solution.

**Combining Color Graph and Composition Matching:** We have reduced *WGMP* into a maximization problem; now, if we

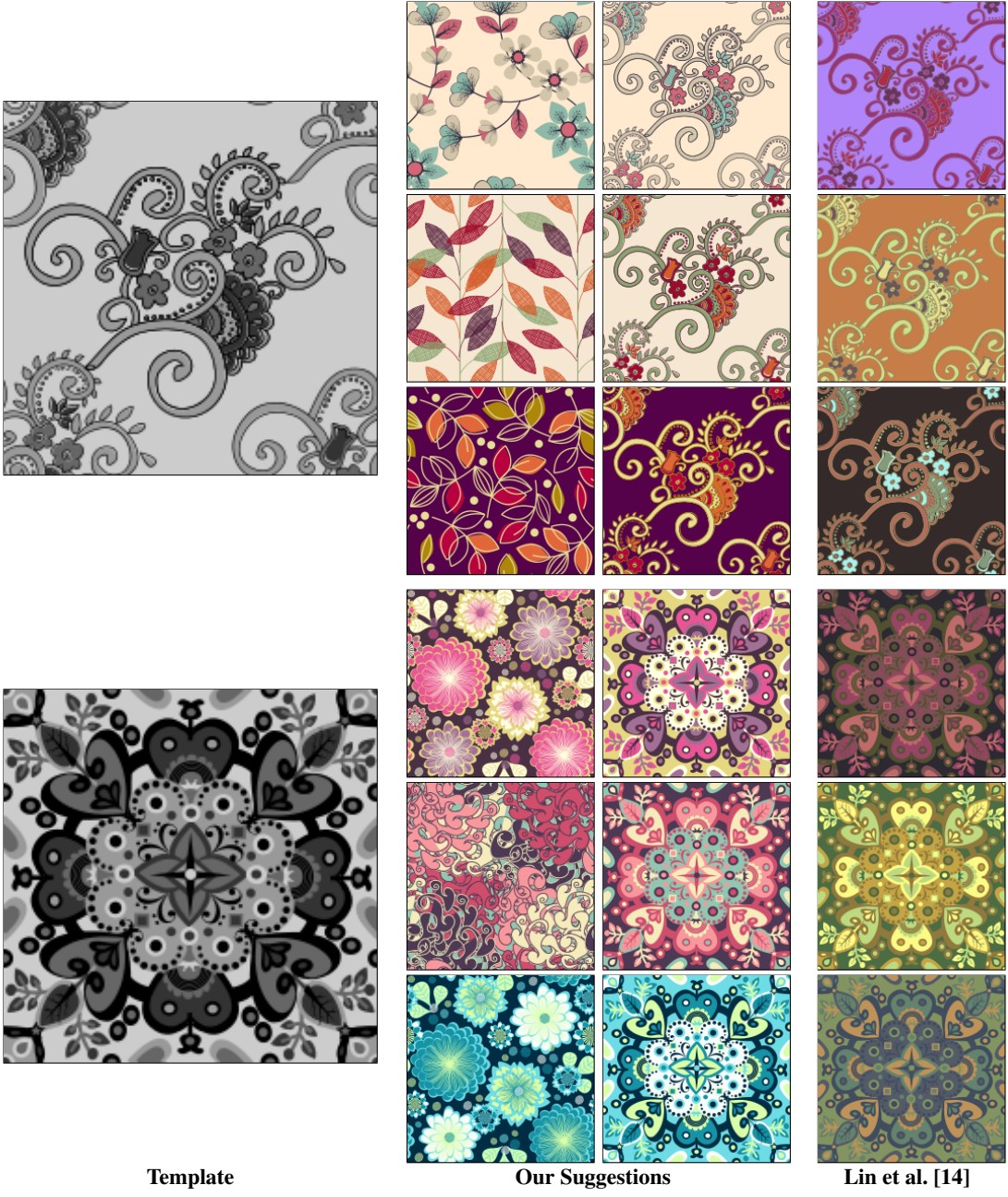

**Template**                          **Our Suggestions**                          **Lin et al. [14]**

Figure 4: Figure shows colorization results for some of the templates (on the left) used for judgement studies. It compare the coloring suggestions generated by our algorithm (in the middle column, left images are the references and right images are the coloring suggestions) with the results of pattern color-by-number (Lin et al. [14] on the right).

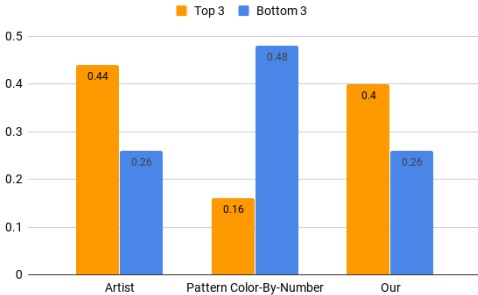

Figure 5: The percentage of times patterns from Artists; our Model and *Pattern-Color-By-Number* were chosen by participants as *Top 3* (favorite) or *Bottom 3* (least favorite) patterns in the judgment study.

negate all the elements of $(\overline{U}_R\overline{U}_T^T)$ in eq. 4, it will become minimization problem. The composition matching is already a minimization problem, so we can add these two cost matrix and define a single matching cost which can be solved for minimization using Hungarian matching algorithm i.e.

$$MatchingCost(M) = M_{cmp} - \overline{U}_R\overline{U}_T^T \qquad (5)$$

$$Matching = \underset{P}{\arg\min}\{tr(P^T M)\} \qquad (6)$$

The permutation matrix $P$ (matching between the color groups) obtained after solving eq. 6, will be used to color each color group of template image with the color of the corresponding matched color group in the reference image.

## 5 EVALUATIONS

**Experiment-1:** Here we compare two fully automatic pattern colorization methods; *Pattern-Color-By-Number* [14] and our algorithm, against the human artists. To conduct this study, we use the same set of pattern templates as used in *Pattern-Color-By-Number* for a similar user study. The set contains a total of 14 templates (one of the 15 templates used in *Pattern-Color-By-Number* is discontinued on the ColourLovers website). The set is reasonably diverse, and templates do not have strong semantic associations. Comparison is done using a human judgement study for different colorization of all 14 gray-scale templates. For each of the 14 templates, we select 12 colorization: four generated using the proposed method, four using *Pattern-Color-By-Number* and four from the top-rated artists (randomly chosen out of 45 top artists for each template) on ColourLovers. For our method, we retrieve 20 reference images from CRID and use 4 best coloring suggestions in terms of aesthetic score (using harmony and contrast similar to [10]). We avoid the random baseline used in *Pattern-Color-By-Number* as users tend to easily identify them.

We recruited 11 masters and Ph.D. students with a normal color vision for the study. We created a Graphical User Interface (GUI), where we display a randomized grid of all 12 colorization for each template. Every participant was asked to judge the *Top 3* (ones they like the most) and the *Bottom 3* (least preferred ones) coloring suggestions. Figure 5 shows the obtained results. We observe that Artist versions are selected most often as a *Top 3* patterns, closely followed by our model. In addition, the coloring suggestions using our models are selected least as a *Bottom 3* pattern (giving similar percentages to the artists versions). Results also indicate the proposed algorithm was significantly preferred over the results generated using *Pattern-Color-By-Number*. Figure 4 contrasts the coloring suggestions generated by both the methods. Our algorithm uses artist created patterns as reference and propagates the colors

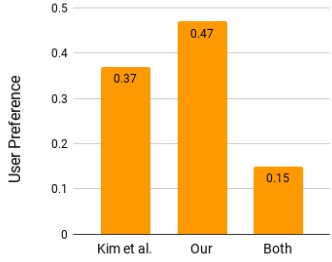

Figure 6: The percentage of times users preferred; our color transfer method; *PERCEPT* and *NotSure* (when no clear preference).

from it to the template. The result suggest that the propagation algorithm is able to preserve favourable aspects from the reference pattern, giving competent results even when compared to purely artist created colorization for a different template.

**Experiment-2:** Here we evaluate our color transfer algorithm against state-of-the-art palette based pattern colorization method *PERCEPT* [10]. The novel aspect of this study is that it only evaluates the color assignment, since both the algorithms use the same palette. A similar perceptual study is conducted by Kim [10] in comparison to *Pattern-Color-by-Numbers*.

We recruited another 10 students with normal vision to conduct this study. The experiments are performed on the same set of source gray-scale templates as used in the previous study. We generate four colorization for each template using our model and four using *PERCEPT*. We re-use the same set of 56 (4 coloring suggestions for each of the 14 unique templates) colorization generated by our model in Experiment 5. We re-compute the color assignments for each of the 56 images using the *PERCEPT* assignment algorithm, keeping the same color palette and the template as used in our model. The versions generated using our model vs *PERCEPT*, are then compared against each other with the same template and palette pair.

We created another GUI for this study where we present a randomly ordered (left and right) grid of two colored patterns for each of the 56 pairs (one pair at a time). User was asked to select the preferred pattern. A 'Not Sure' option was also provided, in case, the user did not have a clear preference among the two. Figure 6 presents the result for this study. It can be observed that patterns coloring produced by our approach are preferred by users over *PERCEPT* by a margin of 10% of the total pairs displayed on GUI. This is an interesting observation, as *PERCEPT* performs an exhaustive search of all possible assignments and evaluates them on aesthetic score, hence the version generated using our approach would be one of the assignments. This indicates that a pure aesthetic prediction model like *PERCEPT*, may not be always preferred by the user. This also suggest that color graph and composition matching proposed in our model, helps to exploit implicit aesthetic components regarding spatial arrangement of colors and provides better color assignments.

## 6 SUMMARY AND CONCLUSION

In this paper, we propose a novel approach for the automatic colorization of gray-scale graphic templates. In contrast to previous approaches, our method tackles both the problems of palette selection and color assignment in a unified framework. The colorization is posed as the propagation of colors from a retrieved artist-colored reference image to input template. The propagation problem is formulated as a color graph and matching between reference and template image. Using two separate judgment studies, we show that the colorization generated using our model are preferred over state of the art approaches Pattern-Color-by-Numbers [14] and PERCEPT [10].

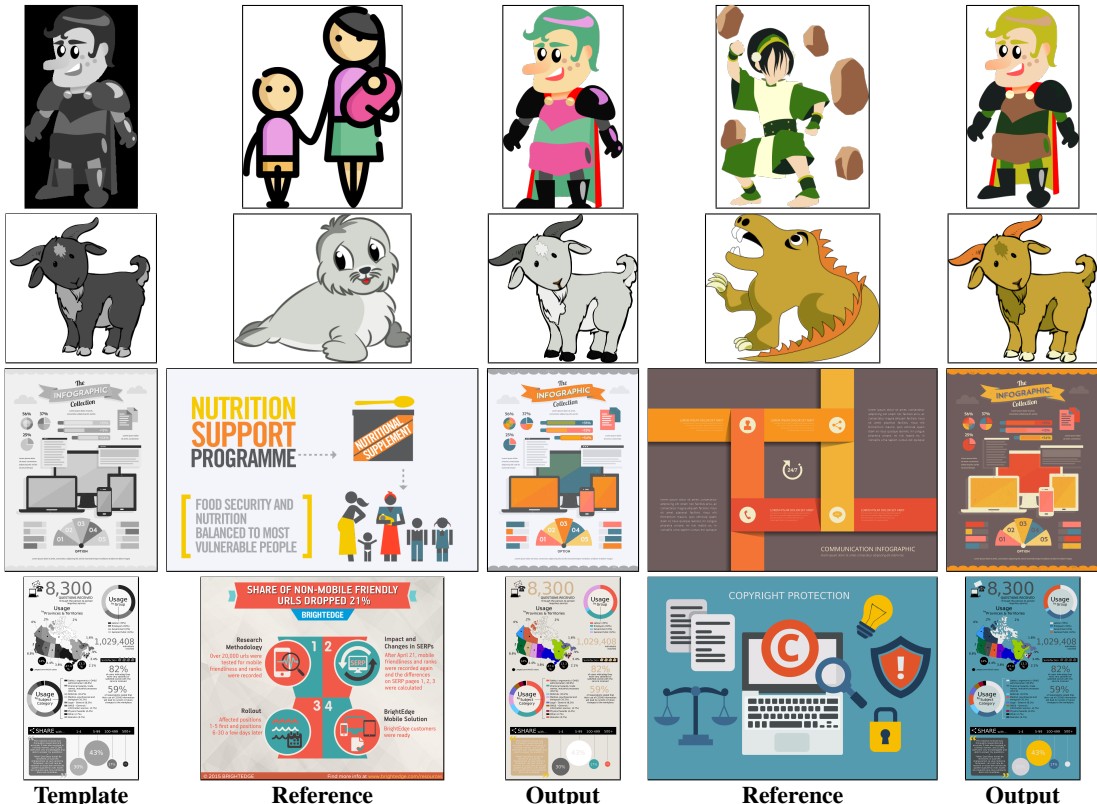

**Template**     **Reference**     **Output**     **Reference**     **Output**

Figure 7: Figure shows different coloring suggestions for the template images on the left. Results demonstrate the utility of proposed colorization into different domains (first two images are cartoon animations and last one is an infographic).

The adaptability of our approach is demonstrated using results on a variety of applications, including graphics arts, infographics and cartoon images. The polynomial-time complexity of our model makes it suitable for coloring the templates, even with a large number of distinct color groups. Figure 7 demonstrates some of the qualitative examples with cartoon templates having only a few colors (second row) to infographics with a large number of distinct colors (row 3). Our model attempts to preserve the original look and feel of the artist by preserving neighborhood and compositions of colors from the reference image. Future work would further explore the adoption of the proposed approach in applications like websites and interior designs and also explore extensions of the framework for colors suggestion for a partially colored template or color suggestions to replace only particular color groups.

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
