# OpenReview forum: "ColorArt: Suggesting Colorizations for Graphic Arts Using Optimal Color-Graph Matching"
_graphicsinterface.org/Graphics_Interface/2020/Conference — GI 2020_

### Official Review · AnonReviewer2 · 2020-04-17
**Review of ColorArt: Suggesting Colorizations For Graphic Arts Using Optimal Color-Graph Matching**

**Rating:** 7
**Confidence:** 3

**Review:**

Paper summary
================================

The paper presents a new method for the automatic colorization of pattern-based images using other colored graphics as a reference. The authors show in a study that their colorization method is superior to others.


References
================================

The references are good.



Implementation
================================

The algorithm is relatively simple and is explained well.



Writing
================================

The writing is good, and easy to follow. Furthermore, the paper is short, and still manages to discuss the method in full details, which is very welcome.

Equations (4) and (6) should specify (as a subscript) over which variable the argmax / argmin is performed.

Minor writing issues:
- p1: one keyword is "G-r-a-phic-arts"
- p1: "It is considered a valuable ingredient of our artistic abilities, so and so, that [...]" - unusual use of "so and so"
- p2: "If the template and reference image has [...]" should be "have"
- p3: "because the content in natural images have" singular / plural mismatch
- p3: "collected from Colourlovers(an online community" space missing between Colourlovers and opening bracket
- p4: "Each element of the vector represents size of corresponding color group" article missing
- p4: "The objective is to propagate colors of reference image to the input template" article missing
- p5: "We created a Graphical User Interface(GUI)" space missing between Interface and opening bracket



Novelty
================================

I am not versed enough in the field of image processing and colorization to judge the novelty of this paper. Given the discussion of previous works presented in this article, this article seems sufficiently novel to warrant publication.



General
================================

This article addresses an interesting problem in image processing with an easy and compelling solution. The authors apply their method to a variety of images to be colored, and perform a user study to quantify their method.

This paper is ready for publication after the minor issues in this review have been addressed.

---

### Official Review · AnonReviewer3 · 2020-04-24
**ColorArt: Suggesting Colorizations For Graphic Arts Using Optimal Color-Graph Matching**

**Rating:** 6
**Confidence:** 3

**Review:**

This paper proposes an interesting approach to colorize grayscale graphics arts by using the color scheme in reference images. Given input templates, it firstly searches similar reference images from a colored image dataset. Then colors can be transferred from reference images to input templates. The colorization pipeline is validated by two different user studies. The results are impressive in the paper and the supplementary materials. But the exposition of the paper should be improved. There are many typos and grammatical errors in the paper, which hinder understanding. I incline to weakly accept the paper if the authors can improve the exposition and address my concerns in the following.

Major issues:

The authors claim that they use an analytic approach other than a combinatorial or an iterative one. But they use the Hungarian algorithm approach to solve the matching problem, which is an iterative method essentially.

The method requires the template and the reference images to have the same number of color groups. It is unclear what are the olor groups distribution in CRID and VG dataset respectively. What are the lower bound and the upper bound of the number of color groups in the implementation based on those datasets?

How to select a reference image from the KNN search from the dataset? Are all k candidates feasible for final results? Are all results in Figure 1 generated using k reference images? What is the value of k in the implementation? It is not mentioned in the paper.

Since the idea of graphic arts colorization is inspired by [5], it would be better to show the results of [5], the network of which can be trained on CRID and VG dataset using a similar way presented in their paper.

Minor issues:

-          Prepositions in the title should not be capitalized;

-          In 4.1, “a reference images are” -> "a reference image is" or "reference images are";

-          In Eq.4, “Pargmax” -> “argmax_P”;

-          In 4.2, the subject is missing after “Composition Matching (Mcmp):”;

-          “eq. eq:maximization” -> eq.4;

-          In Eq.6, “Pargmax” -> “argmax_P”;

-          “eq. eq:minimization” -> eq.6;

-          In Figure 7, “Result demonstrate” -> "Results demonstrate"

---

### Official Review · AnonReviewer1 · 2020-04-24
**Pretty results but not very thorough research**

**Rating:** 5
**Confidence:** 3

**Review:**

The paper idea is pretty much summarized in the title: given a greyscale template image, they find the closest colored reference image in some custom metric, and match the greyscale values to the colors using a weighted graph matching. The technical contribution of the paper is formulating the problem as an instance of graph matching and solving it using a polynomial-time approximation introduced in [22].

The overall idea of the paper is quite clean and straightforward: in order to color a template, it makes sense to find a geometrically similar image and use its colors, matching geometrically similar elements. However, this is exactly where the paper falls a little short on its promise: instead of some notion of geometric similarity, the two out of three features the paper uses to compare images are purely pixel statistics ('Composition' and 'Sum of gradient'). I guess I would have expected some more geometric features like shape context or at least some histograms of gradient.

Moreover, the paper doesn't really demonstrate how well the reference image search works. I would have expected some validation of just that stage: perhaps, a few closest images and a few far images in the collection, at the very least. Otherwise it's hard to judge whether it works at all (Fig. 4 doesn't help here: template doesn't look like the ref image).

Finally, I'm a little surprised to see the discussion of Experiment-2. Do I understand correctly that the authors infer their method is superior to PERCEPT based on 10% out of 10 students' opinions, i.e. one person on average? I do realize it is not a completely formal user study, but I would not call this statistically significant.

So while the paper is written well and clearly, while I appreciate the novelty of formulating the problem as a graph matching and using an approximation algorithm, even while the final colorizations look pretty, the issues above make me a little less optimistic about the paper. If the other reviewers think those issues are minor, I won't argue though.

---

### Meta-Review · Area_Chair1 · 2020-04-24

**Recommendation:** Accept
**Confidence:** 3

**Metareview:**

All the reviewers agree that the results are impressive, and the problem is interesting. The exposition and validation of the paper, however, need some work (R1, R3). Since overall the reviewers are positive about the paper, I recommend accepting it with the following provisions:

- Clarify or rephrase the analysis of the second user study (R1)
- Add missing details on kNN (R3) and, if possible, a figure demonstrating how well the reference search works (R1)
- Fix typos (R2, R3).

---

### Decision · Program_Chairs · 2020-04-25

Accept